# Lifestyle factors associated with pre-metabolic syndrome in young adults: A cross-sectional study of annual health examinations in university students

Haruka Arimori[1,2]*, Akie Moriuchi[3], Masakazu Kobayashi[4], Shimpei Morimoto[5], Seiko Nakamichi[4], Atsushi Kawakami[1], Norio Abiru[1]

1 Department of Endocrinology and Metabolism, Division of Advanced Preventive Medical Sciences, Graduate School of Biomedical Sciences, Nagasaki University, Nagasaki, Japan, 2 Department of Nursing Science, Faculty of Nursing and Nutrition, University of Nagasaki, Nishi-Sonogi District, Nagasaki Prefecture, Japan, 3 Department of Endocrinology and Metabolism, Nagasaki University Hospital, Nagasaki, Japan, 4 Health Center, Nagasaki University, Nagasaki, Japan, 5 Innovation Platform & Office for Precision Medicine, Nagasaki University Graduate School of Biomedical Sciences, Nagasaki, Japan

* jj20230362@ms.nagasaki-u.ac.jp

## Abstract

The recent global rise in obesity and metabolic syndrome (MetS) among young adults has become a public health concern. Moreover, lifestyle changes were widely reported as a result of preventive measures against the novel coronavirus disease 2019. We conducted a cross-sectional study to investigate lifestyle factors associated with a pre-disease condition prior to MetS in young adults during/post-pandemic era. A survey was distributed to fourth-year students at Nagasaki University in 2023, and medical examination data were collected. Participants who met both high intra-abdominal fat area (IAFA >71.1 cm²) and at least one of the Japanese diagnostic criteria for MetS were categorized as having pre-MetS. The pre-MetS group was compared with the control (non–pre-MetS) group to examine its characteristics using Fisher's exact test and binomial logistic regression. A total of 856 students participated in this study; of them, 43 (5.0%) were classified as the pre-MetS group. Fisher's exact test identified significant associations between pre-MetS and breakfast frequency of 2–3 times/week, dining out ≥4 times/week, no or infrequent part-time job, late bedtime, and longer gaming time. Multivariable logistic regression revealed that after mutual adjustment, pre-MetS remained associated with no or infrequent part-time job and longer gaming time. After further adjustment for age, sex, and body mass index, only no or infrequent part-time job and gaming time of ≥4 h/day were significantly associated with pre-MetS. These factors were associated with pre-MetS and might reflect early metabolic alterations; further prospective studies are needed in this regard.

**Data availability statement:** The dataset supporting the findings of this study has been deposited in the figshare repository and is openly accessible: Arimori H, Moriuchi A, Kobayashi M, Morimoto S, Nakamichi S, Kawakami A, Abiru N. MetS_factors_and_life-style_factors_Nagasaki_University_students_2023_4grades.csv. 2005. Figshare Dataset. https://doi.org/10.6084/m9.figshare.29946917.v1.

**Funding:** This work was supported by JSPS KAKENHI, grant number JP21K11575.

**Competing interests:** The authors declare no conflict of interest.

## Introduction

The global rise in obesity among young people has recently become a public health concern [1]. Between 1990 and 2022, obesity rates increased significantly not only among adults but also among school-aged children and adolescents [2]. Metabolic syndrome (MetS) increases the risk of cardiovascular disease and type 2 diabetes, and its early detection and prevention are essential across all age groups, including young adults. Recent studies on the prevalence and associated factors of pre-MetS exist [3,4]; however, they are retrospective or limited to specific countries, making the findings insufficient. Therefore, further research on pre-MetS is warranted.

In Japan, MetS is defined as a waist circumference (WC) of ≥85 cm in males and ≥90 cm in females, corresponding to an intra-abdominal fat area (IAFA) of 100 cm², combined with at least two of the following conditions: (1) high blood pressure (BP) [systolic blood pressure (SBP) ≥130 mmHg and/or diastolic blood pressure (DBP) ≥85 mmHg], (2) dyslipidemia fasting triglycerides (TG) ≥150 mg/dL and/or high-density lipoprotein cholesterol (HDL-c) <40 mg/dL), and (3) impaired fasting plasma glucose ≥110 mg/dL [5].

Our previous study found that 3.3% of male students at Nagasaki University had MetS. Participants were divided into four categories based on their IAFA; <50 cm², 50–74 cm², 75–99 cm², and ≥100 cm². The adjusted odds ratios (ORs) for having ≥2 MetS components compared with the IAFA <50 cm² group were 4.80, 7.34, and 37.56, indicating a dose-dependent positive association between higher IAFA and the presence of multiple MetS components [6]. Furthermore, a study investigating the relationship between IAFA and MetS component development in male university students found a significant increase in BP in participants with IAFA ≥50 cm² [7]. These findings suggest that, in Japanese young adults, an IAFA of ≥50 cm²—even if <100 cm²—may indicate pre-MetS.

A previous study from Korea reported that an IAFA of 71.1 cm² was the optimal cutoff associated with obesity-related metabolic abnormalities in adolescents aged between 16 and 18 years [8]. As this finding was derived from an East Asian population, it may serve as a useful reference when identifying early metabolic alterations among young Japanese adults.

Since December 2019, lifestyle changes such as changes in dietary habits, sleep patterns, and low physical activity have been widely reported as a result of preventive measures against the novel coronavirus disease 2019 (COVID-19) [9]. In this regard, we conducted a survey of Nagasaki University students during the 2020–2021 academic year, which was marked by lifestyle restrictions due to the COVID-19 pandemic confinement. Several lifestyle factors were found to be significantly associated with self-reported weight gain of ≥3 kg during the period from before the mild lockdown in Japan (October–December 2019) to the period in mild lockdown (October–December 2020), including eating out frequency of ≥4 times/week and gaming time of ≥4 h/d among males, and time spent at home of ≥12 h/d among females [10]. With the advent of vaccines and therapeutic agents for COVID-19 [11,12], as well as the acquisition of herd immunity through previous infections, societies have transitioned from the pandemic to the post-COVID-19 era, resulting in additional lifestyle changes.

In this study, we developed an original definition of the pre-MetS in young people based on objective health examination data collected from students during/after the COVID-19 era. We sought to identify individuals who met this definition and investigate novel factors associated with pre-MetS.

## Materials and methods

### Study design and participants

The study participants were recruited during their annual student health examination, which took place between April and June 2023. Eligibility criteria included the ability to understand Japanese and give informed consent, regardless of nationality.

This academic year, 1,441 fourth-year students were enrolled at Nagasaki University, with 1,423 undergoing the student health examination, during which the following data were collected: anthropometric measurements (height, body weight, WC, BP, laboratory analyses), and IAFA using the dual bioelectrical impedance analysis (BIA) instrument (Omron Dual scan HDS-2000; Omron, Kyoto, Japan). Blood was drawn from the antecubital vein, and plasma glucose, TG, and HDL-c levels were determined using standard laboratory methods. IAFA was measured using BIA, which is a practical and non-invasive technique for large-scale examinations [13], although it is less precise than CT and MRI. Indeed, validation studies have shown that BIA tends to underestimate IAFA compared with CT and may exhibit sex-related bias [14]. Therefore, IAFA values should be interpreted with caution. A web-based questionnaire was administered through a custom-designed web form that was accessible *via* smartphones to examine lifestyle-related factors among the participants. The response rate was 59.4% (n = 856; 481 males and 375 females; mean age: 21.8 ± 2.0 years) (Fig 1).

### Definition of pre-disease state of MetS

In this study, pre-MetS in young people was defined as the presence of elevated IAFA and at least one MetS component.

Regarding the IAFA, the cutoff IAFA value was set at 71.1 cm², which was reported to be associated with obesity-related diseases in young adults based on a previous report [8]. The participants' median value was 34.8 cm² (S1 Fig).

Regarding MetS components other than IAFA, the Japanese diagnostic criteria for MetS [5] apply only to fasting biochemical test data; however, in this study, fasting data were available for only 50.8% (n = 435) of the participants. The cutoffs for BP and fasting biochemical test data were adopted from the Japanese diagnostic criteria for MetS, and those for casual biochemical test data were determined with reference to the diagnostic cutoff values for dyslipidemia and diabetes [15,16].

Consequently, in this study, a new criterion was considered to assess pre-MetS, high IAFA (≥71.1 cm²), and at least one of the MetS components—high BP (SBP ≥ 130 mmHg and/or DBP ≥ 85 mmHg), dyslipidemia (fasting TG ≥ 150 mg/dL, non-fasting TG ≥ 175 mg/dL, and/or HDL-c < 40 mg/dL), or hyperglycemia (fasting plasma glucose ≥110 mg/dL or casual plasma glucose ≥200 mg/dL). In this group, the pre-MetS group, participants underwent further analysis (S2 Fig).

Although this definition is based on a combination of an externally validated IAFA cutoff and the Japanese diagnostic criteria for MetS, it remains somewhat arbitrary and not universally recognized. Therefore, this operational definition should be regarded as exploratory rather than a validated diagnostic construct, serving instead as a practical approach to identify young individuals with early metabolic alterations related to MetS.

### Questionnaire survey

Before conducting the survey, participants provided informed consent. Participants were then given a QR code that allowed them to easily access and complete the web-based questionnaire via Google Forms®. The survey was completed conveniently while waiting for the health exam. Notably, it took approximately 5 min to complete. The received data were analyzed with confidentiality in mind.

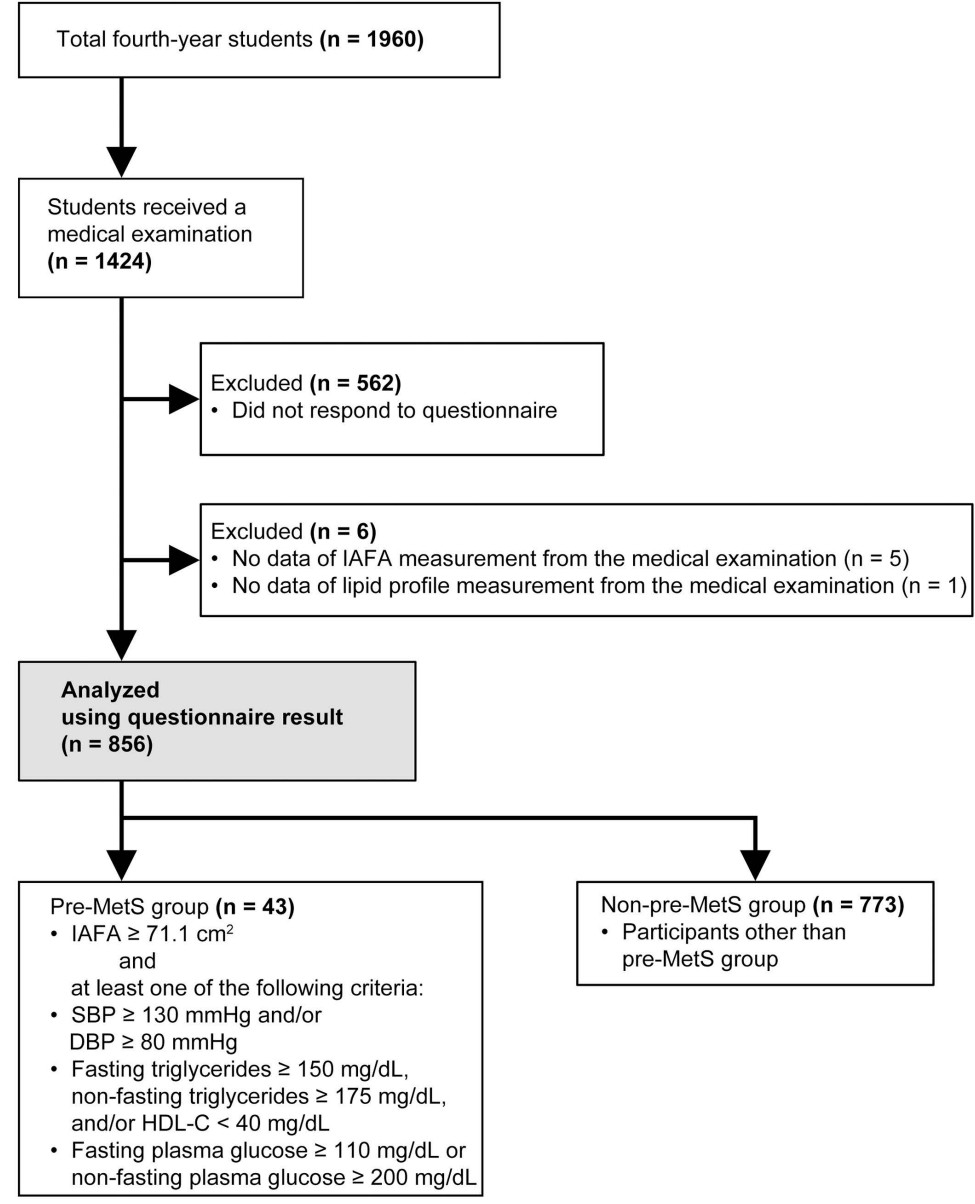

**Fig 1. Study flow chart.** The pre-metabolic syndrome (pre-MetS) group included students who had IAFA ≥71.1 cm² and at least one of the Japanese diagnostic criteria for MetS—high BP (SBP ≥ 130 mmHg and/or DBP ≥ 85 mmHg), dyslipidemia (fasting TG ≥ 150 mg/dL, non-fasting TG ≥ 175 mg/dL, and/or HDL-c < 40 mg/dL), or hyperglycemia (fasting plasma glucose ≥110 mg/dL or casual plasma glucose ≥200 mg/dL). The non–pre-MetS group included students other than those in the pre-MetS group).

The questionnaire, based on the Pittsburgh Sleep Quality Index [17] and the National Health and Nutrition Survey Ministry of Health, Labour and Welfare [18], contained nine sections with a total of 47 items listed as follows:

(1) Personal information (age and sex);

(2) Physical activity (staying at home (h/d), part-time job frequency (times/week), and social club activities frequency (times/week));

(3) Dietary patterns (breakfast (times/week) and dining out (times/week));

(4) Daily routine (bedtime, wake-up time, and sleep duration (h));

(5) Lifestyle (smoking, drinking (times/month), alcohol amount (units/time), gaming time (h/d), and internet surfing time (h/d));

(6) Ten-item internet gaming disorder test (IGDT-10, a score of 5 or more on the IGDT-10 was used as the cutoff) [19];

(7) Generalized Anxiety Disorder-7 (GAD-7) [20];

(8) 6-item Kessler Psychological Distress Scale (K6) [21];

(9) Athens insomnia scale (AIS [22]).

### Identification of lifestyle factors associated with pre-MetS

Categorical variables from the questionnaire, such as lifestyle factors, were divided into three to four categories. The healthiest category or the most participants was chosen as the reference group. For each categorical variable, the associations between lifestyle factors and pre-MetS were found by comparing the pre-MetS group with the non–pre-MetS group.

### Statistical analysis

We used R version 4.0.3, which was developed by the R Development Core Team [23] to conduct statistical analyses. Except for age, which was calculated as a mean value, all continuous variables were presented as mean ± standard deviation (SD). In the IGDT-10 scoring, a score of 0 h of gaming time/d was considered 0 points.

The association between pre-MetS and lifestyle factors was quantified as the ORs and 95% confidence interval. The null hypotheses were rejected at the significance level of 0.05. The significant lifestyle factors identified in this study may represent novel factors associated with pre-MetS in young adults. As such, these findings are considered noteworthy and have been presented in the results section.

### Ethical issues

The ethical review board of Nagasaki University approved the research (approval number: 20062604), which was conducted following the principles of the Declaration of Helsinki. Before the survey, each participant signed an informed consent form.

Because university students are considered a vulnerable population, special ethical considerations were taken to protect them. Participation was entirely voluntary, and participants were reassured that refusal to participate or withdrawal at any time would not affect their academic performance or their relationship with academic advisors. The purpose and methods of the study as well as the anticipated burdens, potential risks and benefits, and intended use of the data were fully explained, and written informed consent was obtained from them after confirming their understanding.

### Results

The pre-MetS group consisted of 43 students (5.0% of the participants), whereas the non–pre-MetS group had 813. The pre-MetS group was significantly older than the non–pre-MetS group with male predominance (90.7% vs. 54.4%, p < 0.001). The pre-MetS group had significantly greater height, body weight, body mass index (BMI), BP, WC, and IAFA than the non–pre-MetS group (p < 0.05), and no significant differences in K6, AIS, IGDT10, and GAD-7 scores were found between the two groups (Table 1).

**Table 1. Characteristics of the survey participants.**

|  | All |  | Pre-MetS group |  | Non–pre-MetS group |  | *p*-value [a] |
|---|---|---|---|---|---|---|---|
| N (male, %) | 856 | (481, 56.2) | 43 | (39, 90.7) | 813 | (442, 54.4) | <0.001** |
| Age (25th–75th percentile) | 21.8 | (21–22) | 22.6 | (21–23) | 21.8 | (21–22) | 0.040* |
| Height (SD), cm [b] | 165.5 | (±8.4) | 169.0 | (±7.6) | 165.3 | (±8.4) | 0.003* |
| Body weight (SD), kg [b] | 57.7 | (±10.6) | 78.2 | (±12.2) | 56.7 | (±9.4) | <0.001** |
| BMI (SD), kg/m² [b] | 21.0 | (±2.9) | 27.4 | (±3.9) | 20.7 | (±2.4) | <0.001** |
| SBP (SD), mmHg [b] | 120.5 | (±11.5) | 131.6 | (±9.4) | 119.9 | (±11.3) | <0.001** |
| DBP (SD), mmHg [b] | 69.2 | (±8.5) | 78.8 | (±9.2) | 68.7 | (±8.1) | <0.001** |
| WC (SD), cm [b] | 71.9 | (±8.3) | 90.9 | (±8.7) | 70.9 | (±7.0) | <0.001** |
| IAFA (SD), cm² [b] | 38.4 | (±24.0) | 100.0 | (±25.7) | 35.1 | (±19.0) | <0.001** |
| K6 (SD) [b] | 2.2 | (±3.6) | 2.6 | (±3.6) | 2.2 | (±3.6) | 0.480 |
| AIS (SD) [b] | 3.4 | (±2.8) | 3.4 | (±2.8) | 3.4 | (±2.8) | 0.960 |
| IGDT-10 (SD) [c] | 1.3 | (±2.2) | 2.0 | (±2.4) | 1.3 | (±2.2) | 0.063 |
| GAD-7 (SD) | 2.8 | (±3.7) | 2.6 | (±3.6) | 2.8 | (±3.7) | 0.712 |
| Plasma glucose (SD) | 85.5 | (±12.3) | 85.7 | (±13.7) | 85.5 | (±12.2) | 0.937 |
| TG (SD) | 89.8 | (±66.8) | 178.0 | (±129.7) | 85.1 | (±58.3) | <0.001** |
| HDL-c (SD) | 63.5 | (±13.8) | 49.1 | (±9.2) | 64.2 | (±13.6) | <0.001** |

AIS, Athens insomnia scale; BMI, body weight (kg)/height (m²); DBP, diastolic blood pressure; SBP, systolic blood pressure; GAD-7, generalized anxiety disoder-7; HDL-c; high-density lipoprotein cholesterol; IAFA, intra-abdominal fat area; IGDT-10, Ten-item internet gaming disorder test; K6, 6-item Kessler psychological distress scale; TG, triglycerides; WC, waist circumference.

Pre-MetS group (pre-metabolic syndrome group): students who had IAFA of ≥71.1 cm² and at least one of the components for MetS — high BP (SBP ≥ 130 mmHg and/or DBP ≥ 85 mmHg), dyslipidemia (fasting TG ≥ 150 mg/dL, non-fasting TG ≥ 175 mg/dL, and/or HDL-c < 40 mg/dL), or hyperglycemia (fasting plasma glucose ≥110 mg/dL or casual plasma glucose ≥200 mg/dL). The non–pre-MetS group included students who were other than those included in the pre-MetS group.

[a]p-value: p-value based on Fisher's exact or Welch's t-tests in the pre-MetS and non–pre-MetS groups. * p < 0.05, ** p < 0.001.

[b]Anthropometric factors (height, body weight, BMI, SBP, DBP, and WC), bioelectrical impedance analysis (IAFA with DUALSCAN), K6 and AIS scores, and biochemical test data were obtained from the physical examination data.

[c]IGDT-10: Students who reported their gaming time = 0 h/d were analyzed with 0 points for their score of IGDT-10.

Fisher's exact test analysis of lifestyle factors in the pre-MetS and non–pre-MetS groups revealed that the following were significantly associated with pre-MetS: having breakfast of 2–3 times/week (OR 0.23, p = 0.030), dining out ≥4 times/week (OR 2.69, p = 0.044), part-time job frequency of 0 (OR 6.62, p < 0.001) and 1–2 (OR 3.87, p = 0.007) times/week, bedtime after 2 am (OR 2.86, p = 0.011), and gaming time <2 (OR 4.45, p = 0.001), 2–4 (OR 6.82, p < 0.001), and ≥4 (OR 10.99, p < 0.001) h/d (Fig 2).

A binary logistic regression analysis of these five factors demonstrated significant associations with the results of the Fisher's exact test. After adjusting for the other four lifestyle factors, pre-MetS was found to be associated with part-time job (frequency of 0 time/week and 1–2 times/week) and gaming time (<2, 2–4, and ≥4 h/d) (Table 2).

Furthermore, after adjusting for sex, age and BMI, pre-MetS was found to be associated with part-time job frequencies of 0 and 1–2 times/week, and gaming time of ≥4 h/d. The associations between gaming time of <2 and 2–4 h/d with pre-MetS remained significant after adjusting for age and sex; however, these associations disappeared after additional adjustment for BMI (Table 3).

## Discussion

### Differences between this study and our previous findings

Unlike our previous pandemic-era survey [10], which relied on self-reported data and found that frequently eating out was associated with weight gain, the current study—based on objectively measured IAFA and biochemical parameters—did

| Category | | OR (95%CI), p value | # (pre-MetS group/ non-pre-MetS group) |
|---|---|---|---|
| **Breakfast frequency (times/week)** | ≥6 | Reference | 19/305 |
| | 4–5 | 0.78 (0.30–2.05), 0.820 | 6/123 |
| | 2–3 | **0.23 (0.04–0.90), 0.030*** | 2/141 |
| | ≤1 | 1.05 (0.51–2.10), 1.000 | 16/244 |
| **Dining out frequency (times/week)** | <1 | Reference | 13/285 |
| | 1 | 1.05 (0.44–2.41), 1.000 | 11/229 |
| | 2–3 | 1.03 (0.43–2.36), 1.000 | 11/234 |
| | ≥4 | **2.69 (1.05–7.04), 0.044*** | 8/65 |
| **Drinking frequency (times/week)** | Non + Ex † | Reference | 10/98 |
| | ≤2 | 0.47 (0.22–1.06), 0.057 | 32/674 |
| | ≥3 | 0.24 (0.01–1.66), 0.183 | 1/41 |
| **Alcohol amount (units/time)** | <2 | Reference | 31/591 |
| | 2–4 | 0.88 (0.37–1.99), 0.847 | 8/173 |
| | ≥4 | 1.56 (0.48–4.61), 0.345 | 4/49 |
| **Smoking status** | Non | Reference | 40/737 |
| | Ex | 0.41 (0.02–6.79), 0.621 | 0/22 |
| | Occasion § | 0.52 (0.03–8.81), 1.000 | 0/17 |
| | Daily ‖ | 1.49 (0.38–4.74), 0.462 | 3/37 |
| **Time spent at home (h/day)** | <8 | Reference | 11/296 |
| | 8–12 | 1.31 (0.55–3.22), 0.531 | 12/247 |
| | ≥12 | 1.99 (0.93–4.35), 0.095 | 20/270 |
| **Part-time job frequency (days/week)** | 0 | **6.62 (2.36–18.83), <0.001**** | 19/183 |
| | 1–2 | **3.87 (1.43–11.16), 0.007*** | 16/264 |
| | 3–4 | Reference | 5/320 |
| | ≥5 | 4.15 (0.84–17.58), 0.074 | 3/46 |
| **Social club activity frequency (days/week)** | ≥4 | 0.38 (0.02–2.28), 0.504 | 1/44 |
| | 2–3 | 0.50 (0.16–1.45), 0.288 | 4/133 |
| | ≤1 | Reference | 38/636 |
| **Bedtime (o'clock)** | ~11 pm | 1.67 (0.25–7.96), 0.627 | 2/31 |
| | 11 pm–0.5 am | Reference | 10/260 |
| | 0.5–2 am | 1.03 (0.44–2.41), 1.000 | 15/377 |
| | 2 am– | **2.86 (1.24–6.66), 0.011*** | 16/145 |
| **Waking up time (o'clock)** | –8 am | Reference | 17/283 |
| | 8–10 am | 0.74 (0.38–1.49), 0.386 | 18/404 |
| | 10 am~ | 1.06 (0.43–2.58), 1.000 | 8/126 |
| **Sleep duration (h/day)** | ~6 | 1.27 (0.64–2.52), 0.478 | 13/202 |
| | 6–9 | Reference | 30/590 |
| | 9~ | 0.45 (0.03–7.61), 0.617 | 0/21 |
| **Gaming time (h/day)** | 0 | Reference | 5/343 |
| | <2 | **4.45 (1.66–12.49), 0.001*** | 21/323 |
| | 2–4 | **6.82 (2.29–20.80), <0.001**** | 11/110 |
| | ≥4 | **10.99 (3.03–38.45), <0.001**** | 6/37 |
| **Internet surfing time (h/day)** | <1 | Reference | 1/26 |
| | 1–2 | 1.27 (0.17–29.82), 1.000 | 6/123 |
| | 2–4 | 0.94 (0.14–20.19), 1.000 | 15/416 |
| | ≥4 | 2.20 (0.36–46.59), 0.705 | 21/248 |

Odds ratio (log scale): 0  1  4  12  24  48

**Fig 2. Relationship between pre-MetS and each lifestyle factor.** * $p < 0.05$; ** $p < 0.01$. † Non: non-drinker or non-smoker, Ex: ex-drinker or ex-smoker. § Occasion, occasional smoker. ‖ Daily, daily smoker. OR, odds ratio. Pre-MetS group (pre-metabolic syndrome group): students who had an IAFA of ≥71.1 cm² and at least one of the components for MetS — high BP (SBP ≥ 130 mmHg and/or DBP ≥ 85 mmHg), dyslipidemia (fasting TG ≥

150 mg/dL, non-fasting TG ≥ 175 mg/dL, and/or HDL-c < 40 mg/dL), or hyperglycemia (fasting plasma glucose ≥ 110 mg/dL or casual plasma glucose ≥ 200 mg/dL). The non-pre-MetS group comprised students who were not included in the pre-MetS group. Error bars indicate 95% confidence intervals.

not find such association. These differences likely reflect not only the post-COVID transition period but also the shift from subjective, self-reported data to objective health examination data collected from fourth-year students.

## Gaming time

In this study, longer gaming time was associated with pre-MetS. Although IGDT-10 scores did not differ between groups, prolonged gaming—even below the threshold of clinical gaming disorder—may lead to increased sedentary time, reduced energy expenditure, and a higher risk of obesity [24,25]. The adverse health effects of sedentary behavior might be independent of the benefits of physical activity [26–28], and prolonged sedentary time might be associated with obesity and MetS that cannot be offset by increased activity [29]. Moreover, increased screen time and sedentary behavior have been linked to unhealthy eating patterns that may contribute to obesity and MetS [27,28,30–33]. Problematic gaming may also affect physical health through sleep-related and other psychiatric disorders [34–40], while sleep deprivation and circadian rhythm disruption themselves have been associated with obesity and liver disease [41,42]. However, these mechanisms remain speculative, and it is possible that gaming time in this study serves as an associated factor of pre-MetS through sedentary behavior rather than as a direct causal factor.

## Frequency of part-time job

Previous studies have found that part-time jobs are associated with increased physical activity among male high school students in the USA [43] and Canada [44]. In this study, pre-MetS was associated with either infrequent or no part-time job. One possible explanation is lower daily activity or higher sedentary time; however, this interpretation is hypothetical and requires validation with objective activity measures.

## Dietary habits

In this study, breakfast frequency and dining out were associated with pre-MetS in univariate analyses, but not after multi-variable adjustment. This attenuation likely reflects confounding or mediation by other lifestyle factors such as BMI, gaming time, and part-time job frequency. Although dietary variables were not independently significant, irregular breakfast consumption and frequent dining out have been consistently linked to obesity and MetS in prior studies [45,46]. Therefore, dietary habits remain important behavioral factors for metabolic health, even if their effects were not independently observed in this young cohort.

## Comparisons with other populations and study implications

Similar associations between physical inactivity, sedentary behavior, unhealthy dietary habits, and early metabolic alterations have been reported in university populations from Western, Middle Eastern, and African regions [47–50]. Compared with these international findings, our study highlights distinctive lifestyle features of Japanese students—particularly prolonged gaming time and limited part-time work—which may reflect underlying patterns of low physical activity and extended sedentary time. Although novelty is modest, this study provides updated, population-specific data on intra-abdominal fat and lifestyle factors among Japanese young adults, contributing contextually relevant evidence to the global understanding of early metabolic health.

## Limitations

This study has several critical limitations. First, the operational definition of pre-MetS (IAFA ≥71.1 cm² plus ≥1 MetS component) was based on an exploratory analysis rather than on an established or validated diagnostic criterion. Therefore,

**Table 2. Binary logistic regression analysis of the five lifestyle factors that showed significant differences in the Fisher's exact test associated with pre-MetS.**

| Lifestyle Factors | Category | Units | Odds Ratio | (95% Confidence Interval) | *p*-value |
|---|---|---|---|---|---|
| # Breakfast frequency and dining out frequency | | | | | |
| Breakfast frequency | 2–3 | times/week | 0.23 | (0.04–0.82) | 0.051 |
| Dining out frequency | ≥4 | times/week | 2.80 | (1.05–7.08) | 0.033* |
| # Breakfast frequency and part-time job frequency | | | | | |
| Breakfast frequency | 2–3 | times/week | 0.24 | (0.04–0.85) | 0.058 |
| **Part-time job frequency** | **0** | **days/week** | **6.32** | **(2.49–19.37)** | **$3.2 \times 10^{-4}$**  |
| | **1–2** | **days/week** | **3.74** | **(1.44–11.57)** | **0.011*** |
| # Breakfast frequency and bedtime | | | | | |
| Breakfast frequency | 2–3 | times/week | 0.25 | (0.04–0.92) | 0.073 |
| Bedtime | After 2 | Am | 3.28 | (1.39–8.07) | 0.007* |
| # Breakfast frequency and gaming time | | | | | |
| Breakfast frequency | 2–3 | times/week | 0.24 | (0.04–0.87) | 0.061 |
| **Gaming time** | **<2** | **h/day** | **4.69** | **(1.87–14.24)** | **0.002*** |
| | **2–4** | **h/day** | **7.93** | **(2.75–26.17)** | **$2.3 \times 10^{-4}$**  |
| | **≥4** | **h/day** | **14.18** | **(3.93–53.51)** | **$4.7 \times 10^{-5}$**  |
| # Dining out frequency and part-time job frequency | | | | | |
| Dining out frequency | ≥4 | times/week | 3.45 | (1.28–8.86) | 0.011* |
| **Part-time job frequency** | **0** | **days/week** | **7.39** | **(2.87–22.86)** | **$1.0 \times 10^{-4}$**  |
| | **1–2** | **days/week** | **3.99** | **(1.53–12.37)** | **0.008*** |
| # Dining out frequency and bedtime | | | | | |
| Dining out frequency | ≥4 | times/week | 2.23 | (0.83–5.67) | 0.097 |
| Bedtime | After 2 | Am | 2.55 | (1.12–6.05) | 0.028* |
| # Dining out frequency and gaming time | | | | | |
| Dining out frequency | ≥4 | times/week | 2.08 | (0.78–5.25) | 0.127 |
| **Gaming time** | **<2** | **h/day** | **4.36** | **(1.75–13.20)** | **0.004*** |
| | **2–4** | **h/day** | **6.43** | **(2.27–20.87)** | **$7.6 \times 10^{-4}$**  |
| | **≥4** | **h/day** | **10.01** | **(2.85–36.62)** | **$2.9 \times 10^{-4}$**  |
| # Part-time job frequency and bedtime | | | | | |
| **Part-time job frequency** | **0** | **days/week** | **7.28** | **(2.84–22.45)** | **$1.2 \times 10^{-4}$**  |
| | **1–2** | **days/week** | **4.10** | **(1.57–12.71)** | **0.007*** |
| Bedtime | After 2 | Am | 3.47 | (1.53–8.23) | 0.003* |
| # Part-time job frequency and gaming time | | | | | |
| **Part-time job frequency** | **0** | **days/week** | **6.73** | **(2.63–20.71)** | **$2.2 \times 10^{-4}$**  |
| | **1–2** | **days/week** | **4.41** | **(1.69–13.70)** | **0.005*** |
| **Gaming time** | **<2** | **h/day** | **4.76** | **(1.90–14.45)** | **0.002*** |
| | **2–4** | **h/day** | **7.39** | **(2.60–24.09)** | **$3.2 \times 10^{-4}$**  |
| | **≥4** | **h/day** | **10.85** | **(3.08–39.93)** | **$1.9 \times 10^{-4}$**  |
| # Bedtime and gaming time | | | | | |
| Bedtime | After 2 | Am | 2.19 | (0.96–5.19) | 0.065 |
| **Gaming time** | **<2** | **h/day** | **4.42** | **(1.75–13.51)** | **0.004*** |
| | **2–4** | **h/day** | **6.31** | **(2.19–20.83)** | **0.001*** |
| | **≥4** | **h/day** | **9.88** | **(2.75–37.14)** | **$4.1 \times 10^{-4}$**  |

* *p* < 0.05; ** *p* < 0.001. We conducted binary logistic regression analysis of these five factors that showed significant differences in the Fisher's exact test associated with dual overlapping components. These factors included breakfast frequency of 2–3 times/week (reference: ≤1 time/week), dining out frequency of ≥4 times/week (reference: <1 time/week), part-time job frequency of 0 days/week and 1–2 days/week (reference: 3–4 days/week), bedtime after 2 am (reference: 11:00 pm–12:30 am), and gaming time of <2 h/d, 2–4 h/d, and ≥4 h (reference: 0 h/d).

**Table 3. Association of part-time job frequency and gaming time with pre-MetS, adjusted for sex, age, and BMI.**

| Lifestyle Factors | Category | Units | Odds Ratio | (95% Confidence Interval) | p-value |
|---|---|---|---|---|---|
| Part-time job frequency (adjusting for sex, age, and BMI) | 0 | days/week | 3.92 | (1.13–15.46) | 0.038* |
| | 1–2 | days/week | 4.88 | (1.52–18.76) | 0.012* |
| Gaming time (adjusted for sex, age, and BMI) | <2 | h/day | 1.11 | (0.32–4.42) | 0.876 |
| | 2–4 | h/day | 1.64 | (0.41–7.26) | 0.493 |
| | ≥4 | h/day | 5.65 | (1.04–31.69) | 0.043* |
| Gaming time (adjusted for sex and age) | <2 | h/day | 3.26 | (1.26–10.30) | 0.024* |
| | 2–4 | h/day | 4.16 | (1.41–14.18) | 0.013* |
| | 4 | h/day | 6.51 | (1.80–24.86) | 0.004* |

* $p < 0.05$; ** $p < 0.001$. Multivariable analysis was conducted for the following factors: part-time job frequency of 0 days/week and 1–2 days/week (reference: 3–4 days/week); gaming time of <2 h/day, 2–4 h/day, and ≥4 h/day (reference: 0 h/day); age; sex; and BMI.

its definition should be interpreted with caution and recognized as a potential limitation. Second, the relatively small number of participants in the pre-MetS group (n = 43) limited the statistical power to strengthen observed associations. In addition, the small sample size restricted adjustments for potential unmeasured confounders; thus, a larger sample would confirm these findings. Third, the sample was drawn from a single cohort of fourth-year students at one Japanese university, with a 59.4% response rate, which may have introduced self-selection and institutional bias. Although participants' BMI and BP were comparable to national student data [51], generalizability remains limited to this homogeneous, academically specific population. Fourth, lifestyle variables such as physical activity, gaming time, alcohol use, and sleep were self-reported and not validated by objective measures (e.g., actigraphy, dietary records), which may have introduced recall and social desirability biases. Fifth, IAFA was assessed using BIA, which can be influenced by hydration and technical factors. While practical for large-scale screening, BIA is less precise than gold-standard imaging methods such as CT or MRI, and the resulting IAFA estimates should be interpreted with caution. Sixth, the observed association between the absence of a part-time job and pre-MetS is speculative and likely reflects lower physical activity, although objective activity data were not collected. Seventh, other potential confounders—such as family history, physical activity intensity, and socioeconomic status—were not considered, possibly limiting interpretability. Finally, the cross-sectional design precludes causal inference. Overall, given these multiple limitations, the findings should be interpreted cautiously, and future studies with larger and more diverse samples and objective assessments are needed to confirm and extend these results.

## Conclusion

In this study, longer gaming time and infrequent or lack of a part-time job were significantly associated with pre-MetS (IAFA >71.1 cm² and at least one of the MetS components) among university students. These lifestyle factors may be associated with an early pre-disease metabolic state related to MetS in young adults and warrant further investigation through prospective studies to clarify possible causal relationships and inform early preventive strategies. From a public health perspective, the present findings suggest the importance of developing campus-level health promotion initiatives—such as programs that raise awareness of sedentary behavior and support balanced daily routines among university students. Although these findings do not imply causality, they may provide useful insights for designing early preventive approaches targeting young adults.

## Supporting information

**S1 Fig. Histogram of Intra-abdominal fat area of participants.**
(TIF)

**S2 Fig. Venn diagram of IAFA and blood pressure, blood glucose, and lipids in 856 participants.** IAFA (intra-abdominal fat area), participants with IAFA ≧71.1 cm$^2$. Blood pressure, participants with SBP≧130 mmHg and/or DBP≧85 mmHg. Lipid, participants with fasting TG>150 mg/dL, non-fasting TG≥175 mg/dL, and/or HDL-c<40 mg/dL. *Includes participants with hyperglycemia (fasting plasma glucose ≥110 mg/dL or casual plasma glucose ≥200 mg/dL). The number following the asterisk indicates the number of participants with hyperglycemia. For example, N=856 (*3) indicates that out of 856 participants, 3 had hyperglycemia. The area enclosed by red dotted lines indicates the pre-MetS group: participants with high IAFA (71.1 cm$^2$) and at least one MetS component (i) SBP≧130 mmHg and/or DBP≧85 mmHg, (ii) fasting TG>150 mg/dL, non-fasting TG≥175 mg/dL, and/or HDL-c<40 mg/dL, and/or (iii) fasting plasma glucose ≥110 mg/dL or casual plasma glucose ≥200 mg/dL). (TIF)

## Acknowledgments

We are grateful to Masaki Miwa, Hiromi Yamamoto, and Chikako Yamaura from the Diabetes Care Support Center, Nagasaki University for their support and assistance in conducting the survey. The authors would also like to thank Enago (www.enago.jp) for the English language review.

## Author contributions

**Conceptualization:** Haruka Arimori, Akie Moriuchi, Masakazu Kobayashi, Norio Abiru.

**Data curation:** Haruka Arimori, Masakazu Kobayashi.

**Formal analysis:** Haruka Arimori, Shimpei Morimoto.

**Funding acquisition:** Masakazu Kobayashi, Atsushi Kawakami.

**Investigation:** Haruka Arimori, Akie Moriuchi.

**Methodology:** Haruka Arimori, Akie Moriuchi, Masakazu Kobayashi, Norio Abiru.

**Project administration:** Masakazu Kobayashi, Norio Abiru.

**Resources:** Masakazu Kobayashi, Seiko Nakamichi.

**Software:** Masakazu Kobayashi.

**Supervision:** Akie Moriuchi, Norio Abiru.

**Validation:** Shimpei Morimoto, Norio Abiru.

**Visualization:** Haruka Arimori.

**Writing – original draft:** Haruka Arimori.

**Writing – review & editing:** Akie Moriuchi, Masakazu Kobayashi, Shimpei Morimoto, Seiko Nakamichi, Atsushi Kawakami, Norio Abiru.

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
