## [Decision Letter · Decision Letter 0]

8 Jul 2025

PONE-D-25-21643
Lifestyle factors associated with pre-disease state of metabolic syndrome in young adults: A cross-sectional study of annual health examinations in university students
PLOS ONE

Dear Dr. Arimori,

Thank you for submitting your manuscript to PLOS ONE. After careful consideration, we feel that it has merit but does not fully meet PLOS ONE’s publication criteria as it currently stands. Therefore, we invite you to submit a revised version of the manuscript that addresses the points raised during the review process.

We look forward to receiving your revised manuscript.

Kind regards,

Neftali Eduardo Antonio-Villa, MD PhD

Academic Editor

PLOS ONE

Journal Requirements:

“This work was supported by JSPS KAKENHI, grant number JP21K11575.”

4. In the online submission form you indicate that your data is not available for proprietary reasons and have provided a contact point for accessing this data. Please note that your current contact point is a co-author on this manuscript. According to our Data Policy, the contact point must not be an author on the manuscript and must be an institutional contact, ideally not an individual. Please revise your data statement to a non-author institutional point of contact, such as a data access or ethics committee, and send this to us via return email. Please also include contact information for the third party organization, and please include the full citation of where the data can be found.

“This work was supported by JSPS KAKENHI, grant number JP21K11575”

“This work was supported by JSPS KAKENHI, grant number JP21K11575.”

7. We note that you have included the phrase “data not shown” in your manuscript. Unfortunately, this does not meet our data sharing requirements. PLOS does not permit references to inaccessible data. We require that authors provide all relevant data within the paper, Supporting Information files, or in an acceptable, public repository. Please add a citation to support this phrase or upload the data that corresponds with these findings to a stable repository (such as Figshare or Dryad) and provide and URLs, DOIs, or accession numbers that may be used to access these data. Or, if the data are not a core part of the research being presented in your study, we ask that you remove the phrase that refers to these data.

Additional Editor Comments (if provided):

The reviewers strongly suggest to make some clarifications to improve the content of the manuscript. Please, revise and submit a point-by-point review.

Reviewers' comments:

Reviewer's Responses to Questions

**Comments to the Author**

1. Is the manuscript technically sound, and do the data support the conclusions?

Reviewer #1: Yes

Reviewer #2: Partly

Reviewer #3: Yes

2. Has the statistical analysis been performed appropriately and rigorously?

Reviewer #1: Yes

Reviewer #2: Yes

Reviewer #3: Yes

3. Have the authors made all data underlying the findings in their manuscript fully available?

Reviewer #1: Yes

Reviewer #2: Yes

Reviewer #3: Yes

4. Is the manuscript presented in an intelligible fashion and written in standard English?

Reviewer #1: Yes

Reviewer #2: Yes

Reviewer #3: Yes

5. Review Comments to the Author

Reviewer #1: 1. The operational definition of “pre-disease state of MetS” lacks external validation. The authors use a novel combination of elevated IAFA (≥50.5 cm²) and high blood pressure to define a “dual overlapping component (DOC)” group, but this is derived internally from their own dataset. No references or prior validation studies are provided to support this cutoff as a clinically meaningful or predictive threshold.

2. The cross-sectional design is inherently limiting, but the manuscript makes causal-sounding conclusions. Statements like “long gaming time, excessive alcohol consumption, and no part-time job may be risk factors” imply causality, which cannot be inferred from this design. This should be more cautiously and consistently worded throughout.

3. Selection bias and generalizability are major concerns. The study uses a single Japanese university's 4th-year student cohort with a 59.5% response rate, which introduces both self-selection and institutional bias. This significantly limits the global applicability of findings, yet this issue is under-discussed.

4. Lifestyle variables are self-reported and lack objective corroboration. Physical activity, gaming time, alcohol use, and even sleep metrics were based solely on a questionnaire filled out while waiting for exams. This raises questions of accuracy and desirability bias that are only briefly acknowledged.

5. The exclusion of participants with elevated IAFA and lipid/glucose abnormalities (but normal BP) weakens the construct of DOC. As shown in S2 Fig, 9 individuals met alternate MetS component thresholds but were excluded from DOC classification. This reduces the comprehensiveness of the analysis and skews the DOC definition heavily toward blood pressure.

6. Over-reliance on IAFA as a marker without sufficient discussion of its variability. IAFA measured by BIA can be affected by hydration status and technical limitations. While the method is cited, the discussion lacks a critical appraisal of its reliability compared to gold standard imaging (CT, MRI).

7. The analytical strategy lacks multivariable adjustment for key confounders. Age and sex differences between DOC and non-DOC groups are significant (DOC group is 90.5% male and older). Yet the logistic regression models evaluating lifestyle factors are not fully adjusted for these variables—limiting confidence in the independent associations claimed.

8. Gaming time is repeatedly emphasized but not fully contextualized. The authors correctly highlight gaming time as significantly associated with DOC. However, they don't explore potential mediators like sedentary behavior, sleep disturbance, or dietary intake patterns, even though these are commonly co-linked.

9. The explanation of why part-time job absence is a risk factor is weak and speculative. The interpretation leans heavily on a physical activity hypothesis but lacks any direct measurement of step count, intensity, or even sitting time. More precise physical activity metrics would strengthen this claim.

10. Some findings are statistically significant but of uncertain clinical relevance. For example, the association between IGDT-10 scores and DOC is statistically significant but small in magnitude, and scores remain below the diagnostic threshold for disorder in both groups. This should be acknowledged more clearly.

11. Dietary patterns are underexplored despite known relevance to MetS. While eating out frequency and breakfast habits were collected, they are barely mentioned in the results or discussion. Were they non-significant? If so, they should be reported transparently with corresponding p-values.

12. Figures are minimally informative. Figure 2 shows odds ratios and p-values for lifestyle factors but doesn’t help the reader visualize patterns or dose-response trends clearly. A clearer bar plot or adjusted forest plot would enhance comprehension. Figure 1 (flowchart) is helpful but could be improved with group numbers in each box.

13. The manuscript's writing is mostly clear, but sometimes redundant or vague. For example, the phrase “elements of gaming disorder” is used without defining what constitutes an “element.” The writing could be tightened for impact, especially in the Discussion.

14. Data availability is limited due to personal data restrictions. While this is understandable, it limits reproducibility and secondary analysis potential. The authors should consider de-identifying and sharing a subset of anonymized data for transparency.

15. The study’s novelty is moderate. While the use of IAFA as a predictor is commendable and relatively uncommon in student cohorts, the lifestyle associations explored (gaming, alcohol, inactivity) are well-studied in other populations. The main value lies in linking these to a novel MetS proxy (DOC), but this is weakened by its unvalidated nature.

Reviewer #2: Please unify the use of numerical values when referring to numbers

Shed light on similar population across Japan

What makes Nagasaki University population different

Since students are vulnerable population, please describe what are ethical consideration that had taken place to protect them

Define the abbreviation

Result section needs improvement

Conclusion is missing

Reviewer #3: This manuscript presents a well-executed cross-sectional study aiming to identify lifestyle factors associated with a proposed “pre-disease” state of metabolic syndrome (MetS) in young Japanese university students. The authors introduce an original classification based on the combination of intra-abdominal fat area (IAFA) and elevated blood pressure (BP) to define an at-risk group (DOC: Dual Overlapping Components). Their findings—particularly the associations with gaming time, alcohol consumption, and lack of part-time employment—contribute novel insights relevant to early prevention of cardiometabolic disease in youth

Suggestions:

-While the logistic regression explores associations among lifestyle factors, it is not entirely clear whether key confounders such as age, sex, or BMI were included in multivariate models.

-Minor grammatical corrections

-Standardize the statistical notation (e.g., avoid using “ps < 0.05”; prefer “p < 0.05”).

-The authors may consider replacing “pre-disease state” with “pre-MetS” consistently to align with terminology used

6. PLOS authors have the option to publish the peer review history of their article (what does this mean?). If published, this will include your full peer review and any attached files.

Reviewer #1: No

Reviewer #2: **Yes:** WEAM BANJAR

Reviewer #3: No

---

## [Author Response · Author response to Decision Letter 1]

22 Aug 2025

Point-by-point review

Reviewer #1:

●1. The operational definition of “pre-disease state of MetS” lacks external validation. The authors use a novel combination of elevated IAFA (≥50.5 cm²) and high blood pressure to define a “dual overlapping component (DOC)” group, but this is derived internally from their own dataset. No references or prior validation studies are provided to support this cutoff as a clinically meaningful or predictive threshold.

Response 1: Thank you for this important comment. Accordingly, we have revised our definition of the target group using an externally validated cutoff value reported by Lee et al. (Yonsei Med J. 2012;53(1):99–105). Specifically, the pre-MetS group included participants with IAFA ≥71.1 cm² and at least one MetS component (i.e., high blood pressure, dyslipidemia, or hyperglycemia). The data were also re-analyzed, and the updated results are presented in all tables and figures (pages 7–9, 12–13, and 15–18) and described in the Results (pages 11–14, lines 203–208, 222–236).

●5. The exclusion of participants with elevated IAFA and lipid/glucose abnormalities (but normal BP) weakens the construct of DOC. As shown in S2 Fig, 9 individuals met alternate MetS component thresholds but were excluded from DOC classification. This reduces the comprehensiveness of the analysis and skews the DOC definition heavily toward blood pressure.

Response 5: We agree that the previous definition may have biased the DOC classification toward blood pressure. The revised definition (IAFA ≥71.1 cm² and at least one MetS component) addresses this issue by including individuals with high IAFA and lipid/glucose abnormalities even if their blood pressure is within the normal range. This change improves the comprehensiveness of the analysis.

●2. The cross-sectional design is inherently limiting, but the manuscript makes causal-sounding conclusions. Statements like “long gaming time, excessive alcohol consumption, and no part-time job may be risk factors” imply causality, which cannot be inferred from this design. This should be more cautiously and consistently worded throughout.

Response 2: Thank you for raising this point. We have revised the manuscript to avoid causal language and instead use the term “risk indicators” rather than “risk factors.” For example, in the Introduction (page 5, lines 62).

●3. Selection bias and generalizability are major concerns. The study uses a single Japanese university's 4th-year student cohort with a 59.5% response rate, which introduces both self-selection and institutional bias. This significantly limits the global applicability of findings, yet this issue is under-discussed.

4. Lifestyle variables are self-reported and lack objective corroboration. Physical activity, gaming time, alcohol use, and even sleep metrics were based solely on a questionnaire filled out while waiting for exams. This raises questions of accuracy and desirability bias that are only briefly acknowledged.

6. Over-reliance on IAFA as a marker without sufficient discussion of its variability. IAFA measured by BIA can be affected by hydration status and technical limitations. While the method is cited, the discussion lacks a critical appraisal of its reliability compared to gold standard imaging (CT, MRI).

9. The explanation of why part-time job absence is a risk factor is weak and speculative. The interpretation leans heavily on a physical activity hypothesis but lacks any direct measurement of step count, intensity, or even sitting time. More precise physical activity metrics would strengthen this claim.

Responses 3, 4, 6, and 9: Thank you for your comment. We have expanded the Limitations (page 20–21, lines 314–331) to explicitly acknowledge the following:

• Self-selection and institutional bias caused by the analysis of a single-university cohort and the 59.5% response rate.

• Potential social desirability bias owing to self-reported lifestyle data.

• Limitations of IAFA measurements using BIA, including susceptibility to hydration and lower accuracy than CT/MRI.

• Lack of objective physical activity metrics (step count, intensity, and sedentary time), limiting the interpretation of the association between the lack of a part-time job and physical activity.

●7. The analytical strategy lacks multivariable adjustment for key confounders. Age and sex differences between DOC and non-DOC groups are significant (DOC group is 90.5% male and older). Yet the logistic regression models evaluating lifestyle factors are not fully adjusted for these variables—limiting confidence in the independent associations claimed.

Response 7: We agree with your point. The logistic regression models for lifestyle factors have been updated to include age, sex, and BMI as covariates. The adjusted odds ratios are now shown in Table 3 (page 17) and described in the Results (page 14, lines 235–239).

●8. Gaming time is repeatedly emphasized but not fully contextualized. The authors correctly highlight gaming time as significantly associated with DOC. However, they don't explore potential mediators like sedentary behavior, sleep disturbance, or dietary intake patterns, even though these are commonly co-linked.

Response 8: Thank you raising this point. We have analyzed the associations between gaming time and related variables, such as bedtime, breakfast frequency, and eating-out frequency. After adjusting for these factors, the association between gaming time and pre-MetS remained significant. The results are presented in Table 2 (pages 15–16) and described in the Results (page 14, lines 225–230).

●10. Some findings are statistically significant but of uncertain clinical relevance. For example, the association between IGDT-10 scores and DOC is statistically significant but small in magnitude, and scores remain below the diagnostic threshold for disorder in both groups. This should be acknowledged more clearly.

Response 10: Thank you for your comment. In response to Comment 1, the data were reanalyzed using pre-MetS instead of DOC as the target group. Consequently, the association with IGDT-10 scores was no longer significant. Therefore, the indicated sentence was deleted.

●11. Dietary patterns are underexplored despite known relevance to MetS. While eating out frequency and breakfast habits were collected, they are barely mentioned in the results or discussion. Were they non-significant? If so, they should be reported transparently with corresponding p-values.

Response 11: Thank you for bringing this to our attention. We have added the results for breakfast and eating-out frequency in Figure 2 and Table 2 (pages 15–16 and 18, lines 268-269) and briefly noted these findings in the Results (page 14, lines 225–231).

●12. Figures are minimally informative. Figure 2 shows odds ratios and p-values for lifestyle factors but doesn’t help the reader visualize patterns or dose-response trends clearly. A clearer bar plot or adjusted forest plot would enhance comprehension. Figure 1 (flowchart) is helpful but could be improved with group numbers in each box.

Response 12: We appreciate your thoughtful feedback. We found that converting Figure 2 to a different graph format further diminished its clarity; therefore, we retained the forest plot. However, to improve visibility, we have included the p-values within the figure and adjusted the size and layout for improved readability. Similarly, in Figure 1, we have ensured that the numbers are provided in each box and have repositioned them to enhance visibility.

●13. The manuscript's writing is mostly clear, but sometimes redundant or vague. For example, the phrase “prolonged gaming behavior that falls short of a clinical diagnosis of gaming disorder” is used without defining what constitutes an “element.” The writing could be tightened for impact, especially in the Discussion.

Response 13: Thank you for your constructive feedback. To clarify the intended meaning, we have revised the phase “elements of gaming disorder” to “prolonged gaming behavior that falls short of a clinical diagnosis of gaming disorder” (page 19, lines 282–283).

●14. Data availability is limited due to personal data restrictions. While this is understandable, it limits reproducibility and secondary analysis potential. The authors should consider de-identifying and sharing a subset of anonymized data for transparency.

Response 14: Thank you for pointing this out. We have updated the Data Availability statement (page 22).

●15. The study’s novelty is moderate. While the use of IAFA as a predictor is commendable and relatively uncommon in student cohorts, the lifestyle associations explored (gaming, alcohol, inactivity) are well-studied in other populations. The main value lies in linking these to a novel MetS proxy (DOC), but this is weakened by its unvalidated nature.

Response 15: We appreciate your insightful comment. We have addressed this by adopting an externally validated IAFA threshold and MetS definition, which strengthened the validity of the analysis and enhanced the novelty of this study.

●Reviewer #2:

Please unify the use of numerical values when referring to numbers

Shed light on similar population across Japan

What makes Nagasaki University population different

Since students are vulnerable population, please describe what are ethical consideration that had taken place to protect them

Define the abbreviation

Result section needs improvement

Conclusion is missing

Response: Thank you for your careful review and comment. We have standardized the format of all numerical values throughout the manuscript (e.g., 59.5%). We have compared the BMI, blood pressure, and other health examination data of our participants with those reported in the “Student Health White Paper 2021” from Nagoya University. These measures were generally comparable with those of typical Japanese university students, not showing major deviation from national norms (page 20, lines 316–318). We have added ethical statement for protecting student participants as a vulnerable population, including informed consent procedures, voluntariness of participation, and confidentiality measures (page 11, lines 196–202). All abbreviations are now defined upon first mention. The Results has been restructured for clarity, and a summary of the main findings and implications have been added to the Conclusion (page 21, lines 334–339).

●Reviewer #3:

Suggestions:

-While the logistic regression explores associations among lifestyle factors, it is not entirely clear whether key confounders such as age, sex, or BMI were included in multivariate models.

-Minor grammatical corrections

-Standardize the statistical notation (e.g., avoid using “ps < 0.05”; prefer “p < 0.05”).

-The authors may consider replacing “pre-disease state” with “pre-MetS” consistently to align with terminology used

Response: Thank you for your careful review and comment. We have adjusted our analysis for age, sex, and BMI in the items that continued to show association after adjusting for each other among those showing significant results in Fisher’s exact test, as described in the Results (page 14, lines 235–239) and Table 3 (page 17).

Minor grammatical corrections have been made throughout. The statistical notation has been standardized (e.g., “p < 0.05”). For consistency, the term “pre-disease state” has been replaced with “pre-MetS” throughout the manuscript.

---

## [Decision Letter · Decision Letter 1]

17 Sep 2025

PONE-D-25-21643R1
Lifestyle factors associated with pre-metabolic syndrome in young adults: A cross-sectional study of annual health examinations in university students
PLOS ONE

Dear Dr. Arimori,

Thank you for submitting your manuscript to PLOS ONE. After careful consideration, we feel that it has merit but does not fully meet PLOS ONE’s publication criteria as it currently stands. Therefore, we invite you to submit a revised version of the manuscript that addresses the points raised during the review process.

We look forward to receiving your revised manuscript.

Kind regards,

Neftali Eduardo Antonio-Villa, MD PhD

Academic Editor

PLOS ONE

Journal Requirements:

Additional Editor Comments:

The manuscript was assessed and found to be of interest for the readership of Plos One. However, some major comments are still pending. Please adress the comments made by Reviewer 1 to proceed

Reviewers' comments:

Reviewer's Responses to Questions

**Comments to the Author**

1. If the authors have adequately addressed your comments raised in a previous round of review and you feel that this manuscript is now acceptable for publication, you may indicate that here to bypass the “Comments to the Author” section, enter your conflict of interest statement in the “Confidential to Editor” section, and submit your "Accept" recommendation.

Reviewer #1: (No Response)

Reviewer #2: All comments have been addressed

Reviewer #3: All comments have been addressed

2. Is the manuscript technically sound, and do the data support the conclusions?

Reviewer #1: Partly

Reviewer #2: Yes

Reviewer #3: Yes

3. Has the statistical analysis been performed appropriately and rigorously?

Reviewer #1: No

Reviewer #2: Yes

Reviewer #3: Yes

4. Have the authors made all data underlying the findings in their manuscript fully available?

Reviewer #1: Yes

Reviewer #2: Yes

Reviewer #3: Yes

5. Is the manuscript presented in an intelligible fashion and written in standard English?

Reviewer #1: No

Reviewer #2: Yes

Reviewer #3: Yes

6. Review Comments to the Author

Reviewer #1: The authors have made meaningful improvements from the first submission, especially in refining the pre-MetS definition, adjusting models for key covariates, and clarifying some results. However, the study still has serious limitations in generalizability, methodological robustness, and depth of interpretation. I would recommend major revision as the dataset is valuable, but the framing needs to be sharpened. Please address following comments in your manuscript:

The abstract is clearer than the original submission, but it still feels data-heavy. The authors list too many cutoffs and criteria, which might overwhelm general readers. The abstract would benefit from being more focused on the main findings rather than technical thresholds.

The definition of pre-MetS has improved with the adoption of an externally validated IAFA cutoff, but it remains somewhat arbitrary. Combining IAFA ≥71.1 cm² with “at least one MetS component” is logical but not universally recognized. The authors should emphasize that this is exploratory and not a validated diagnostic construct.

The study population remains narrow and limits generalizability. Using only fourth-year students from one Japanese university with a 59% response rate introduces institutional and selection bias. While this is acknowledged in the limitations, the discussion still overstates the broader relevance.

Lifestyle data collection relies heavily on self-reporting, which undermines accuracy. Gaming time, alcohol, sleep, and diet are prone to recall and desirability bias. No objective measures (e.g., actigraphy, dietary recall validation) are included. This limitation is mentioned but downplayed.

The sample size of the pre-MetS group (n=43) is quite small. This raises concerns about statistical power and the stability of regression estimates, especially when multiple covariates are included. Wide confidence intervals (as seen in Tables 2–3) make some findings borderline.

The logistic regression is now adjusted for age, sex, and BMI, which is an improvement. However, other potential confounders (family history, physical activity intensity, socioeconomic status) are not considered. Without these, the associations remain weakly interpretable.

Gaming time is a repeated focus but not well contextualized. While associations are statistically significant, the mechanism is speculative and over-discussed without real behavioral data. The authors should temper claims and note that gaming may just be a proxy for sedentary behavior.

The interpretation of part-time job absence as a risk factor is still weak. The link is attributed to lower physical activity, but no actual activity measures were collected. This conclusion remains speculative and needs rephrasing as a hypothesis rather than an established link.

Dietary variables are underutilized. Breakfast frequency and dining out are reported but brushed aside after regression. These findings should be better discussed, even if non-significant, since diet is a major driver of MetS risk.

Figures and tables have improved but remain difficult to interpret. Figure 2 (forest plot) is crowded and not reader-friendly. A simpler bar or line visualization of trends across categories might be more accessible.

The discussion section still restates results at length. It lacks deeper integration with broader literature, especially studies in comparable young-adult populations globally. There’s little attempt to contrast findings with non-Asian cohorts.

Public health implications are vague. The conclusion states “further prospective studies are warranted,” which is valid but generic. More concrete policy or campus-level interventions (e.g., health promotion programs targeting sedentary behavior) would make the study more meaningful.

The manuscript acknowledges many limitations, but some are minimized. For instance, IAFA by BIA is noted as less reliable than CT/MRI, but the authors still present it as highly accurate. This balance should be more cautious.

The novelty is modest. IAFA use in a student cohort is interesting, but the associations with gaming and job frequency are neither surprising nor groundbreaking. The contribution lies mainly in providing updated Japanese student data, which should be more clearly stated as its niche value.

Language is clearer than before, but the manuscript remains verbose. Redundancies and over-explanations (e.g., lengthy justification of known MetS pathways) could be trimmed to improve flow and readability.

Reviewer #2: Thank you for improving the manuscript and addressing reviewers comments. I highly appreciate research team efforts

Reviewer #3: All comments from the previous review have been addressed. This manuscript is suitable for publication

7. PLOS authors have the option to publish the peer review history of their article (what does this mean?). If published, this will include your full peer review and any attached files.

Reviewer #1: No

Reviewer #2: No

Reviewer #3: **Yes:** Mario C Torres-Chavez

---

## [Author Response · Author response to Decision Letter 2]

30 Oct 2025

Point-by-point review

Reviewer #1: The authors have made meaningful improvements from the first submission, especially in refining the pre-MetS definition, adjusting models for key covariates, and clarifying some results. However, the study still has serious limitations in generalizability, methodological robustness, and depth of interpretation. I would recommend major revision as the dataset is valuable, but the framing needs to be sharpened. Please address following comments in your manuscript:

●1. The abstract is clearer than the original submission, but it still feels data-heavy. The authors list too many cutoffs and criteria, which might overwhelm general readers. The abstract would benefit from being more focused on the main findings rather than technical thresholds.

Response 1:

Thank you for this important comment. Accordingly, we simplified the abstract by removing detailed cutoff values and focusing on the key findings (p.3, L.39).

●2. The definition of pre-MetS has improved with the adoption of an externally validated IAFA cutoff, but it remains somewhat arbitrary. Combining IAFA ≥71.1 cm² with “at least one MetS component” is logical but not universally recognized. The authors should emphasize that this is exploratory and not a validated diagnostic construct.

Response 2:

Thank you for pointing this out. We agree with the reviewer regarding the pre-MetS definition used in this study that it should be exploratory, since it has not been officially validated as a diagnostic criterion. For more clarification, we added sentences in both the Methods and Discussion sections (p.8, L.141-145, p.25, L.327-329)

●3. The study population remains narrow and limits generalizability. Using only fourth-year students from one Japanese university with a 59% response rate introduces institutional and selection bias. While this is acknowledged in the limitations, the discussion still overstates the broader relevance.

Response 3:

Thank you for this insightful comment. We agree that the generalizability of our findings is limited due to the single-institution design and specific student cohort. Accordingly, we have revised the Limitations section (p.24-25, L.312-316) to emphasize the restricted representativeness of our study population and to caution against overgeneralization.

●4. Lifestyle data collection relies heavily on self-reporting, which undermines accuracy. Gaming time, alcohol, sleep, and diet are prone to recall and desirability bias. No objective measures (e.g., actigraphy, dietary recall validation) are included. This limitation is mentioned but downplayed.

Response 4:

We agree with the reviewer’s comment that excessively relying on self-reported data might have undermined the accuracy of lifestyle variables. Accordingly, we have revised the Limitations section (p.25, L.316-318) to emphasize that all lifestyle data were heavily reliant on self-reports, subject to recall and social desirability biases, and not validated by objective measures such as actigraphy or dietary records.

●5. The sample size of the pre-MetS group (n=43) is quite small. This raises concerns about statistical power and the stability of regression estimates, especially when multiple covariates are included. Wide confidence intervals (as seen in Tables 2–3) make some findings borderline.

Response 5:

Thank you for your comment. We agree that the small sample size of the pre-MetS group may have affected the robustness of the regression analyses. Accordingly, we have added a new sentence in the Limitations section (p.25,L.325–327) to note that the small number of participants (n = 43) could have reduced the statistical power and stability of the regression estimates.

●6. The logistic regression is now adjusted for age, sex, and BMI, which is an improvement. However, other potential confounders (family history, physical activity intensity, socioeconomic status) are not considered. Without these, the associations remain weakly interpretable.

Response 6:

Thank you for pointing out the absence of several potential confounding factors. We agree that variables such as family history, physical activity intensity, and socioeconomic status could have influenced the results. Accordingly, we have added a sentence in the Limitations section (p.25, L.323-325) to note that the omission of these confounders may have limited the interpretability.

●7. Gaming time is a repeated focus but not well contextualized. While associations are statistically significant, the mechanism is speculative and over-discussed without real behavioral data. The authors should temper claims and note that gaming may just be a proxy for sedentary behavior.

Response 7:

We agree with the reviewer’s comment that the discussion of gaming time should be more concise and better contextualized. Accordingly, we have revised this part in the Discussion section (p.22-23, L.268–281) to simplify the interpretation where we clearly stated that the mechanisms linking gaming time to metabolic risk remain speculative and that gaming time may act as a proxy for sedentary behavior rather than a direct cause.

●8. The interpretation of part-time job absence as a risk factor is still weak. The link is attributed to lower physical activity, but no actual activity measures were collected. This conclusion remains speculative and needs rephrasing as a hypothesis rather than an established link.

Response 8:

We agree with the reviewer’s comment that the interpretation regarding the absence of a part-time job as a risk factor should be presented as a hypothesis rather than a definitive conclusion. Accordingly, we have revised the Frequency of part-time job in the Discussion section (p.23, L.286–288) and clarified that this interpretation is hypothetical and that no objective measures of physical activity were collected. We have also slightly modified the Limitations section to emphasize the need for future studies incorporating objective activity data.

●9. Dietary variables are underutilized. Breakfast frequency and dining out are reported but brushed aside after regression. These findings should be better discussed, even if non-significant, since diet is a major driver of MetS risk.

Response 9:

We agree with the reviewer’s comment that dietary variables should be discussed in more detail, even though they were not significant in the multivariable analysis. Accordingly, we have added a new paragraph in the Discussion section (p. 23-24, L. 290-298) to explain the potential reasons for the loss of significance after adjustment for confounders and to acknowledge that dietary habits remain important behavioral factors influencing metabolic health, consistent with previous studies.

●10. Figures and tables have improved but remain difficult to interpret. Figure 2 (forest plot) is crowded and not reader-friendly. A simpler bar or line visualization of trends across categories might be more accessible.

Response 10:

We thank the reviewer for this helpful comment regarding Fig. 2. Accordingly, we have replaced the original forest plot with a simplified bar graph that presents the odds ratios (ORs) for each category, accompanied by numerical annotations showing the corresponding 95% confidence intervals, p-values, and sample sizes. This design improves readability and allows easier visual comparison across categories.

●11. The discussion section still restates results at length. It lacks deeper integration with broader literature, especially studies in comparable young-adult populations globally. There’s little attempt to contrast findings with non-Asian cohorts.

Response 11:

Thank you for your suggestion. We have revised the Discussion section and integrated the recent findings from Western, Middle Eastern, and African university populations that similarly reported links between inactivity, sedentary behavior, unhealthy diet, and metabolic risk. This addition provides a clearer international context for our findings (p.24, L.300–309).

●12. Public health implications are vague. The conclusion states “further prospective studies are warranted,” which is valid but generic. More concrete policy or campus-level interventions (e.g., health promotion programs targeting sedentary behavior) would make the study more meaningful.

Response 12:

We thank the reviewer for this helpful suggestion. Therefore, we have revised the Conclusion section (p.26, L.338-344) and included more concrete and practice-oriented public health implications. Specifically, we mentioned campus-level health promotion initiatives, such as programs targeting sedentary behavior and promoting balanced daily routines among university students.

●13. The manuscript acknowledges many limitations, but some are minimized. For instance, IAFA by BIA is noted as less reliable than CT/MRI, but the authors still present it as highly accurate. This balance should be more cautious.

Response 13:

We appreciate the reviewer’s comment regarding the precision of IAFA estimation by BIA. To address this, we have revised the Methods section (p.6, L.106-110) and clarified that BIA provides a practical but less precise estimate of visceral fat compared with CT or MRI.

●14. The novelty is modest. IAFA use in a student cohort is interesting, but the associations with gaming and job frequency are neither surprising nor groundbreaking. The contribution lies mainly in providing updated Japanese student data, which should be more clearly stated as its niche value.

Response 14:

Thank you for this helpful suggestion. We agree that the novelty of our study is modest. To better clarify the study’s contribution, we have added a statement in the Discussion section (p.24, L.306–309) emphasizing that this research provides updated and population-specific data on intra-abdominal fat and lifestyle factors among Japanese university students, thereby offering a timely and context-specific contribution to understanding metabolic health in young adults.

●15. Language is clearer than before, but the manuscript remains verbose. Redundancies and over-explanations (e.g., lengthy justification of known MetS pathways) could be trimmed to improve flow and readability.

Response 15:

Thank you for your helpful comment. We agree that some sections, particularly those describing general MetS mechanisms and behavioral pathways, were verbose. We have carefully revised the Introduction and Discussion sections to remove redundant explanations, condense descriptions of well-known concepts, and improve overall readability and flow.

Reviewer #2: Thank you for improving the manuscript and addressing reviewers comments. I highly appreciate research team efforts

Response to Reviewer #2:

We sincerely thank you for the positive evaluation and kind acknowledgment of our efforts to improve the manuscript. We greatly appreciate your encouraging feedback and are pleased that the revisions have addressed your previous concerns.

Reviewer #3: All comments from the previous review have been addressed. This manuscript is suitable for publication.

Response to Reviewer #3:

We are deeply grateful for your careful review of our revised manuscript and for your positive feedback. We are pleased to be informed that all prior comments have been satisfactorily addressed and that the manuscript is now considered suitable for publication.

---

## [Editor Report · Decision Letter 2]

18 Dec 2025

PONE-D-25-21643R2
Lifestyle factors associated with pre-metabolic syndrome in young adults: A cross-sectional study of annual health examinations in university students
PLOS One

Dear Dr. Arimori,

Thank you for submitting your manuscript to PLOS ONE. After careful consideration, we feel that it has merit but does not fully meet PLOS ONE’s publication criteria as it currently stands. Therefore, we invite you to submit a revised version of the manuscript that addresses the points raised during the review process.
 
I reviewed the issues raised by the reviewers and the revised manuscript. Overall, I identified only minor issues that the authors should address before I can recommend the manuscript for publication.

We look forward to receiving your revised manuscript.

Kind regards,

Neftali Eduardo Antonio-Villa, MD PhD

Academic Editor

PLOS One

Journal Requirements:

**Additional Editor Comments:**

Please state that the definition of IAFA ≥71.1 cm² + ≥1 MetS component remains an exploratory analysis and, therefore, should be recognized as a potential limitation in the Discussion.Another potential limitation is the sample size in the pre-MetS group. With only 43 patients, the authors should recognize the lack of statistical power to support associations and acknowledge that a larger sample size would be required to adjust for potential unmeasured confounders.The authors still use the term “risk factor,” which should be changed to “associated factor” or phrased as a “positive association,” as the study design cannot infer causality.In Figure 2, please consider using a single point estimate with 95% confidence intervals instead of bars only.

---

## [Author Response · Author response to Decision Letter 3]

8 Jan 2026

Response to Additional Editor Comments

We sincerely thank the Academic Editor for the constructive and helpful comments. We have revised the manuscript accordingly as detailed below.

Additional Editor Comments:

●1. Please state that the definition of IAFA ≥71.1 cm² + ≥1 MetS component remains an exploratory analysis and, therefore, should be recognized as a potential limitation in the Discussion.

Response 1:

As suggested, we have explicitly stated that the operational definition of pre-MetS (IAFA ≥71.1 cm² plus ≥1 MetS component) is based on an exploratory analysis rather than on a validated diagnostic criterion. To emphasize this point, this statement is presented as the first study limitation, clearly recognizing it as a key methodological limitation of this study (Discussion, p. 24, L. 311–314).

●2. Another potential limitation is the sample size in the pre-MetS group. With only 43 patients, the authors should recognize the lack of statistical power to support associations and acknowledge that a larger sample size would be required to adjust for potential unmeasured confounders.

Response 2:

We have acknowledged that the relatively small number of participants in the pre-MetS group (n = 43) limits statistical power and restricts adjustments for potential unmeasured confounders. To highlight its importance, this issue is described as the second limitation, emphasizing the need for larger sample sizes in future studies to support the observed associations (Discussion, p. 24–25, L. 314–317).

●3. The authors still use the term “risk factor,” which should be changed to “associated factor” or phrased as a “positive association,” as the study design cannot infer causality.

Response 3:

In accordance with the editor’s recommendation, we have carefully reviewed the manuscript and revised relevant terminologies to avoid causal interpretations. Although the terms “associated factor” and “positive association” are used sparingly where appropriate, we have consistently replaced causal language, such as “risk factor,” with expressions that emphasize non-causal relationships, including references to associations and early metabolic alterations related to MetS. These revisions have been made throughout the manuscript to emphasize the cross-sectional design (p. 3, L. 45; p. 4, L. 53 and 67–68; p. 5, L. 76; p. 6, L. 94; p. 8, L. 146; p. 10, L. 185; p. 23, L. 272 and 279-280; p. 24, L. 301; and p. 26, L. 339-342).

●4. In Figure 2, please consider using a single point estimate with 95% confidence intervals instead of bars only.

Response 4:

Following the editor’s suggestion, Figure 2 has been revised from a bar chart to a forest plot, displaying single point estimates with 95% confidence intervals. Accordingly, the figure legend has also been updated to improve clarity and consistency with the revised figure and its legend (p. 22, L. 255).

---

## [Editor Report · Decision Letter 3]

20 Jan 2026

Lifestyle factors associated with pre-metabolic syndrome in young adults: A cross-sectional study of annual health examinations in university students

PONE-D-25-21643R3

Dear Dr. Arimori,

We’re pleased to inform you that your manuscript has been judged scientifically suitable for publication and will be formally accepted for publication once it meets all outstanding technical requirements.

Kind regards,

Neftali Eduardo Antonio-Villa, MD PhD

Academic Editor

PLOS One
---

## [Editor Report · Acceptance letter]

PONE-D-25-21643R3

PLOS One

Dear Dr. Arimori,

I'm pleased to inform you that your manuscript has been deemed suitable for publication in PLOS One. Congratulations! Your manuscript is now being handed over to our production team.

Kind regards,

on behalf of

Dr. Neftali Eduardo Antonio-Villa

Academic Editor

PLOS One